# Using the Dynamic SWOT Analysis to Assess Options for Implementing the HB-HTA Model

**DOI:** 10.3390/ijerph19127281

**Published:** 2022-06-14

**Authors:** Barbara Więckowska, Monika Raulinajtys-Grzybek, Katarzyna Byszek

**Affiliations:** 1Department of Social Insurance, Warsaw School of Economics, 02-554 Warszawa, Poland; bawie@sgh.waw.pl; 2Department of Management Accounting, Warsaw School of Economics, 02-554 Warszawa, Poland; mrauli@sgh.waw.pl

**Keywords:** health technology assessment, hospital-based HTA, SWOT analysis

## Abstract

This paper is aimed at exploring the role of the HB-HTA ecosystem as an important pathway for popularizing the implementation of innovations in healthcare organizations. The scientific debate has largely been focused on the rising importance of HB-HTA and the principles guiding the process. Solutions implemented by individual countries differ, which may be rooted in historical, cultural, and institutional differences. Our understanding of the impact of individual countries’ healthcare systems on HB-HTA solutions and infrastructure still lacks a basis in interpretative studies. A conceptual framework is proposed to assess the aptness of the HB-HTA model designed for hospitals operating in a country or region, focused on the concepts of adaptiveness and responsiveness to features of the healthcare system present there. A tool is proposed for investigating factors that are likely to assist the successful implementation of the HB-HTA ecosystem. A dynamic SWOT analysis on the case of the HB-HTA model designed for Poland provides interesting insights into the building of the conceptual framework. The results of this study help explain how to create an HB-HTA model that is best adapted to the regional or national healthcare system, including potential risks and opportunities.

## 1. Introduction

Health Technology Assessment (HTA) ensures that decisions regarding technology are based on scientific evidence; therefore, it is at the heart of health policy making. The rationale for introducing the health technology assessment at the hospital level—Hospital-Based HTA (HB-HTA)—is that introducing innovative technologies and evaluating their effectiveness at all stages of the technologies’ life cycle, as well as understanding the consequences of their obsolescence, requires adjustments and measurements at the hospital level [1]. HB-HTA could facilitate bringing together evidence and relevant and reliable information for hospital managers and other decision makers and lead them to good investment decisions [2].

The overarching principle of HB-HTA is to shed light on clinical and economic evidence obtained through solid processes and tools to enable knowledge-based decisions about innovative technologies [3,4]. This is crucial when existing data on return on investment (ROI) are scant and data on effectiveness are unclear or only a part of the marketing strategy of a technology provider [5]. Researchers have also shown that despite the fact that there is a substantial increase in the use of hospital-based health technology assessment, the results arising from HB-HTA are not commonly shared [6]. This could hamper the dissemination of desirable outcomes achieved due to the introduction of the HB-HTA model [7]. A case study from Kazakhstan proved that HB-HTA could also contribute to eliminating inefficient technologies that are being reimbursed in hospital care [8].

HB-HTA provides context-specific and methodologically sound analysis [9]. HB-HTA should include the integration of the following principles: providing synthetic information for hospital decision makers; aiming to define leadership and partnerships; establishing a strategy for HB-HTA models, including HB-HTA units; and targeting the economic aspect in order to allocate adequate resources that ensure the proper operation of the HB-HTA model within a health system [2,10,11].

Assessing and monitoring, which entails the measurement of the short- and long-term impacts of an HB-HTA model, is important in order to evaluate its overall performance, including costs and benefits [12]. The model’s functionality influences the implementation of innovative technologies at hospital facilities, and consequently, can have an impact on deciding to sustain the innovative technology in the hospital, and to scale up its use in other hospitals or even worldwide [13,14,15]. Little is known about tools that can be used to assess HB-HTA models.

This article aims to contribute to the policy debate on effective models of Hospital-Based Health Technology Assessment. The construct of the HB-HTA model at the regional or country level is deeply related to how the healthcare system is built in all its aspects—organization, management, financing, etc. However, an understanding of the role of the healthcare system as a determinant of how to successfully implement HB-HTA is still lacking in theoretical foundations. In the authors’ opinion, an evidence-based understanding of the existing strengths and challenges, as well as risks and opportunities of regional and national healthcare systems is a key prerequisite for designing an optimal HB-HTA model.

This paper introduces a dynamic assessment approach for answering these research questions and contributes to the discussion in the literature regarding implementations of HB-HTA and future directions for development, including new regulations [16,17,18,19,20,21]. Firstly, a conceptual framework is proposed to assess the features of an HB-HTA model designed for a country or region focusing on the concepts of adaptiveness and responsiveness to features of the healthcare system present there. This concept was derived from the framework established by Bezzi [22]. Secondly, a Dynamic Strengths, Weaknesses, Opportunities, and Threats (SWOT) analysis framework is used to predict the results of the implementation of models in public policy [23], in this case, the implementation of HB-HTA in response to the dynamics of the healthcare system. The Dynamic SWOT framework provides evidence of the interdependency between the strengths and weaknesses of the HB-HTA model and the opportunities and threats resulting from the system-based conditions.

As a result, three research questions were formulated to assess the HB-HTA model:How does a specific HB-HTA model designed for a country/region fit into the good practices of HB-HTA as described in the literature?How does a specific HB-HTA model respond to the risks associated with the regional or national healthcare system in which the HB-HTA is to be performed?How does a model adapt to continuous changes in the healthcare system?

This research was built around these questions. The HB-HTA model that we assessed is presented in Section 2. The model was first described in reference to the AdHopHTA guiding principles, which indicate the most important parameters of the HB-HTA model. The assessment was performed using the Dynamic SWOT framework, which is presented in Section 2. The presentation of the results is followed by a discussion of the necessary dimensions of the assessment of the HB-HTA model.

The Dynamic SWOT analysis applied to the case of the HB-HTA model designed for Poland provides interesting insights into the building of a conceptual framework for analyzing HB-HTA models designed for a specific region with respect to endogenous and exogenous factors. The Polish HB-HTA model is used as an example to present the usefulness of the Dynamic SWOT approach. It can be performed to assess any HB-HTA model designed for hospitals in other countries or regions. These findings may help in designing an optimal HB-HTA model, ensuring the effective implementation of the Health Technology Assessment process at the hospital level and incorporating potential opportunities and threats associated with the regional healthcare system.

This research was conducted in 2021 as part of the project “HB-HTA-PL”, funded by the Polish National Center for Research and Development (2019–2022).

## 2. Materials and Methods

The research was designed to find factors that facilitate or impede the implementation of a particular HB-HTA model in a specific region-based context—including both factors that are related to the design of the model and specific to the healthcare system conditions, as well as changes in the HB-HTA model that would ensure its more successful integration and existence in the region-based context.

The HB-HTA model designed for Poland by Gałązka-Sobotka and Kowalska-Bobko [24] was used as an exemplary case. The details of the model are presented later in this section. The model was developed with reference to the AdHopHTA guiding principles framework [2], which characterizes the features that an HB-HTA model should include. This framework is presented later in this section. This research is organized into two steps.

In the first step, the features of the model are described with reference to the AdHopHTA framework [2], and identified as strengths, weaknesses, opportunities, and threats to mark areas in which the model is consistent with the guiding principles. This step of the research made it possible to present how well the proposed model fits into the AdHopHTA framework and how it responds to the local system-based context. Endogenous factors are model-related, whereas exogenous factors are related to the environment to which the HB-HTA model is exposed, namely the region-based healthcare system.

In the second stage of the research, the Dynamic SWOT analysis was applied [23]. The framework of the Dynamic SWOT is presented later in this section. It was used to explore the relationship between endogenous attributes—i.e., the model’s strengths and weaknesses—and exogenous attributes—i.e., the opportunities and threats resulting from the healthcare system.

### 2.1. Case Study: HB-HTA Model Designed for Poland

Typically, the HB-HTA process is performed by innovative hospitals, willing to adopt a new technology. However, due to the high level of regulation that may be present in the healthcare system, as well as the public financing involved in paying for new technologies, sometimes other actors may also be involved. These are often public authorities, serving as regulators or auditors. They take responsibility for some tasks in the process, with the aim of achieving standardization, control, and approval.

The healthcare system in Poland is based on common health insurance. Funds are distributed to healthcare organizations by the National Health Fund, which is a third-party payer contracting providers for publicly funded health services. The prices of health services are homogenous at the country level, and in terms of in-patient services, they are mostly DRG based. The majority of the providers in the hospital sector are public hospitals, which can be owned by regional authorities, the Ministry of Health, or universities. All citizens are entitled to receive publicly funded health services, provided that those services are included in the “basket of health services”. One of the conditions for including the health service in “the basket” is the positive approval of the health technology by the HTA Agency.

The model designed for Poland involves several groups of stakeholders, with the primary role being taken by hospitals, while supplementary decision-maker roles are played by regional branches of central authorities (16 Regional Competence Centers, RCC, in Poland), the public payer (National Health Fund, NHF), and the HTA Agency. Each of those actors has specified tasks (see Figure 1 and Table 1).

The process of HB-HTA is initiated by a hospital that is keen to introduce an innovative health technology. The hospital’s team prepares the HB-HTA report, which includes an assessment of the technology. The report is then reviewed by the Regional Competence Center, based on the methodology developed and updated by the HTA Agency and using regional health information collected and shared by the National Health Fund in a dedicated database.

The results of the review are passed back to the hospital’s management, which makes the final decision about the implementation of the technology. In the case of a positive decision, the hospital may start searching for the funds necessary to implement the technology (e.g., EU funds or owner’s capital), as well as potentially initiating the process of covering the technology with public funding. The process is finished with the provision of a prospective analysis of the HB-HTA process, which is uploaded to the database to serve as feedback for future HB-HTA projects.

The hospital is the creator of the HB-HTA report. The main role of the Regional Competence Center is to coordinate the process at the regional level as well as to review the model and support the hospital’s decision regarding implementation. The National Health Fund is responsible for maintaining the database and updating it with the information needed to create the HB-HTA report. Its responsibility is also the promotion of HB-HTA among hospitals and the provision of training for hospital-based HB-HTA teams. The HTA Agency is responsible for formulating the HB-HTA methodology and report template. Additionally, the HTA Agency is to certify hospitals with the capacity to perform HB-HTA.

### 2.2. Guiding Principles—The AdHopHTA Framework

In this paper, the concept of a functioning model of HB-HTA was adopted from the AdHopHTA project, which describes good practices for HB-HTA [2]. The AdHopHTA guiding principles for HB-HTA include 4 main dimensions: (i) the assessment process; (ii) leadership, strategy, and partnerships; (iii) resources; and (iv) impact; with 15 detailed guidelines (Table 2). The authors of this article adopted the guiding principles from AdHopHTA with one modification. AdHopHTA guiding principle no 13 (financial resources) was rephrased as ‘Sufficient Resources (Financial Included)’, as, in the authors’ opinion, other resources—such as technological, infrastructural or capital—might also be relevant and should be taken into consideration.

AdHopHTA ranks the guiding principles from the most to the least important in an efficient and effective HB-HTA process. Nine recommendations are indicated as being critical for HB-HTA implementation:no. 4 (mission, vision, values, and management),no. 13 (sufficient resources),no. 5 (leadership and communication policy),no. 6 (selection and prioritization criteria),no. 1 (HB-HTA report: scope, hospital context, information requirements),no. 2 (HB-HTA report: methods, tools, and transferability of results),no. 3 (HB-HTA process: independent, unencumbered, transparent, with cooperation with stakeholders and communication),no. 12 (qualified human resources and their development), andno. 10 (cooperation with HTA organizations).

Building an HB-HTA model that does not address these principles properly might lead to performing the HB-HTA process in an inefficient and/or ineffective manner.

### 2.3. The Dynamic SWOT Analysis Framework

The Dynamic SWOT matrix used to evaluate the HB-HTA model in a particular healthcare system context was adapted from the dynamic SWOT analysis proposed by Bezzi [22]. The dynamism and mutual relationship of elements in the SWOT are captured throughout the analysis of pairs of variables included in any of the SWOT parts. The score assigned to each of the pairs depicts the relationship and impact between them.

The scores were assigned by the authors, each of whom first assigned scores individually, before conducting a discussion to form a final consensus assessment. The values given are presented as an integer in the range <−2;2>, and the following meaning was assigned to each of the values:−2—the element in the row is strongly hindered by the element in the column and as a consequence, its impact is fully or significantly reduced;−1—the row element is hindered by the column element, resulting in reduced impact,0—the two elements have no impact on each other;1—the row element is strengthened by the column element, and as a consequence, it has an increased impact;2—the row element is significantly strengthened by the column element, resulting in its having a considerably stronger impact.

Both rows and columns present all of the factors identified in the SWOT in one of four categories—strengths, weaknesses, opportunities, or threats (S, W, O, T). Factors described as strengths (S) and weaknesses (W) are endogenous factors, related to the construct of the HB-HTA model, whereas opportunities (O) and threats (T) are exogenous factors, resulting from the healthcare system in which the HB-HTA will operate. Rows present dependent variables, the impact of which is affected by the relationship with factors in the columns. The total for each row shows the influence, which is the absolute relative importance of a given factor, and is dependent on endogenous and exogenous circumstances (see Figure 2).

The analysis per column presents all factors—both endogenous and exogenous—as independent variables. They have an impact on other variables in the model, and the total per column is dependent on the absolute strength and direction of the influence that a given factor has—so the columns present the interdependence.

All factors in the SWOT analysis are simultaneously dependent and independent variables, which reflects the dynamics of the model. The model enables one to draw two types of conclusions, which help analyze any HB-HTA model chosen for analysis:Factors that most significantly shape the model in the given conditions and their character (endo- versus exogenous), as well as their influence (positive or negative),Moderators that most significantly affect the model and their quality and degree of impact—strengthening vs. hindering, strong or mild.

A Dynamic SWOT analysis can provide a more rational, evidence-based understanding of the conditions that matter for the implementation of the HB-HTA model in a given healthcare system, as well as their interdependence. The results of the Dynamic SWOT analysis can also be the starting point for correcting the model as a way to support the impact of the strengthening moderators and exploiting the positive factors shaping the models if their influence was hindered in the primary proposal.

The total Dynamic SWOT matrix can be divided into four fields, each presenting a different area of the analysis. The first field (SW-SW) shows the Characteristics of the model, with an emphasis on how endogenous factors affect each other. This helps to identify any possible cannibalism effect and internal incoherence of the HB-HTA model. The field representing the impact of exogenous factors (OT-OT) shows the dynamics of the Environment, namely in what way the healthcare system is evolving. The field in which endogenous factors are influenced by the exogenous ones (SW-OT) shows the Adaptiveness of the model. This can be interpreted as describing whether the model is influenced by external factors and how flexible it is. The last field (OT-SW) shows the Responsiveness of the model, which represents to what extent the model responds to the challenges in the healthcare system.

## 3. Results

### 3.1. HB-HTA Model Designed for Poland via AdHopHTA Guiding Principles

Firstly, the case HB-HTA model was characterized using the AdHopHTA guiding principles. Nearly 100 descriptive parameters were determined by the authors and grouped according to the guiding principles. Next, they were categorized as strengths, weaknesses, opportunities, or threats. The results of this step are presented in Figure 3, which allows the initial identification of the areas that are the model’s strong and weak points, as well as those that serve as chances or risks to the model. At this stage, a simplifying assumption was made to treat all parameters as being equally important to the model.

The highest number of parameters were categorized as strengths, and represent mainly the area of the HB-HTA process (no. 3). The weaknesses of the model are mainly related to skilled human resources and career development (no. 12). Opportunities were identified mainly regarding the HB-HTA report (no. 2) as well as mission, vision and values, and governance (no. 4), and interestingly, most threats were also linked to this guiding principle (no. 4).

Taking into consideration the list of AdHopHTA guiding principles indicated as critical for the HB-HTA implementation it should be noted that the model’s weaknesses and threats were identified in terms of two of them (no. 4 and no. 12). This serves as a justification for a deeper, qualitative analysis.

### 3.2. The SWOT Analysis of the Polish HB-HTA Model

All the descriptive parameters of the HB-HTA model were grouped into a concise SWOT analysis (Table 3). The important features are identified in each part of the table.

Strengths result from the four most important characteristics of the model: centrally developed methodology that is uniform for all participants of the process; a professionally run knowledge base containing a repository of reports, along with procedures for disseminating knowledge; reviews being held by the employees of a regional government institution that is not involved in the process of acquiring and delivering health services; and the independence of the hospital in its final decision making.

Among the weaknesses, the first concerns several organizational and procedural gaps in the description of the model. The remaining three weaknesses result from the decentralization of the process among several public institutions and the lack of overall coordination. The process is dispersed between institutions, which creates the need to develop resources in all of them, additionally increasing the risk of communication problems and the duplication of some activities. Additionally, the model lacks centrally conducted coordination and comprehensive supervision.

The existence of the legal and institutional framework of the health technology assessment process (both HTA and HB-HTA) increases the importance of this process, makes it part of the country’s health policy, and formalizes the procedures of its performance. The better-educated and more aware personnel involved in various institutions also present an opportunity for the development of the HB-HTA. Several tools for popularizing knowledge about HB-HTA and identifying good practices are embedded in the model, and constitute a flywheel for its further popularization and the inclusion of new hospitals. The last opportunity is related to the consistency of the HB-HTA concept with the existing healthcare paradigm, both in Poland and globally.

The environment may pose a threat to the development of the model. The primary source of this may be the resistance of hospitals to the introduction of changes and implementation of external regulations in the field of HB-HTA. This process may no longer be perceived as being fully steered by the hospital. The monopolization of the HB-HTA process by public structures may lead to inhibitory actions on the part of the stakeholders currently conducting HB-HTA activities. Another threat results from the imperfect and highly diversified method of collecting the data necessary for analyses. Information gaps, as well as the varied quality and format of data, may hinder the efficient flow of information within HB-HTA between the hospital and external institutions. The last threat results from a lack of support, both financial and institutional, for the development of HB-HTA, on the part of the government administration as well as on the part of individual actors within the model, e.g., due to other emergent issues such as the COVID-19 pandemic.

### 3.3. Dynamic SWOT Analysis of the Polish HB-HTA Model

In this section of the study, it will be indicated how the individual parameters of the model affect each other, and which of them affect the final success of the implementation and functioning of the HB-HTA model.

Following the Dynamic SWOT methodology, the impact was determined for each pair of factors as an integer within the range <−2;2>. Items in columns are treated as independent variables (impact) and items in rows are treated as dependent variables (response) (Table 4).

When comparing the impact of the individual SWOT elements on the analyzed HB-HTA model, it should be stated that the highest impact on the model can be assigned to two weaknesses (W1 and W4), and the following external factors: T4, followed by O1 and O4 (Figure 4).

The analysis of the rows in Table 4 shows which variables are the most susceptible to the impact. The importance of two strengths (S2 and S3) is hindered by both endogenous and exogenous factors. The only strength that is reinforced by the model is S1. Additionally, all weaknesses in the model are reinforced.

In the case of the external factors, the impact is ambiguous—some of the factors are reinforced and others are hindered. The model works in favor of improving managerial competencies in hospitals and other institutions (O2). Additionally, the threat of data transmission and quality problems (T3) is significantly limited. On the other hand, the model significantly strengthens the threat associated with the resistance of hospitals (T1).

The results of the Dynamic SWOT analysis can be presented in four dimensions: characterization, adaptiveness, responsiveness, and business environment (see Figure 2). The analysis of the absolute impact within the area of characterization indicated that the model’s weaknesses (assumed values from 2 to 9 points) are much more significantly influenced than the strengths of the model (values from 0 to 4 points), and that the most important internal element of the model—gaining 9 points, which is almost twice as much as the other characteristics—was organizational and procedural vagueness (W1). This hinders all strengths identified in the model (see Figure 5).

## 4. Discussion

A successful assessment of the HB-HTA model should be performed in the following dimensions [2]: (1) characteristics of the HB-HTA model; (2) the environment in which it operates; (3) the adaptiveness of the model to the environment; and (4) the responsiveness of the model to challenges that occur. The Dynamic SWOT analysis facilitates a comprehensive assessment of the HB-HTA model, and can provide information on its performance and the actions that need to be taken in order to improve the function of the model.

There are difficulties in assessing and comparing the characteristics of HB-HTA models due to the scarcity of data [13,25]. HB-HTA, similarly to HTA, cannot solve problems with implementation innovation itself; the major challenge is to develop a policy structure that can support the diffusion of the technology [26]. A country-level model based on the personnel and material resources existing in regulatory institutions can support this process; however, this requires additional resources, which may cause a problem [26]. HB-HTA models coordinated by a central agency can yield information about the use of technologies, their compatibility, and the future demands and needs of healthcare.

Given the great differences between national health systems, it is necessary to develop HB-HTA guidelines, manuals, and toolkits suitable for national conditions, with reference to the experience of foreigners in this area. Because hospitals have their own set of unique characteristics, it is necessary to prepare a nationally and internationally recognized HB-HTA guidance manual and toolkit for hospitals that includes, for example, their sizes and specializations, or the specific roles that hospitals may play in the country’s health system, and then to adjust the pilot evaluation process to these characteristics [27]. Hospitals need a practical and contextualized assessment of the use of a specific clinical procedure, medical device, or equipment in relevant settings [11].

The HB-HTA model designed for Poland transfers some activities to several public governmental agencies, which is in line with the idea of a formalized coordination. However, the results of the Dynamic SWOT analysis show that the model’s weaknesses have a much greater impact than the strengths of the model. For instance, numerous sources of unclarity (W1) in areas of responsibility as well as with respect to the exact role of each of the governmental agencies—local branches of central authorities (16 Regional Competence Centers), the public payer (National Health Fund), and the HTA Agency—significantly hinder all of the model’s strengths.

The lack of definition in the means of constructing the knowledge base and the procedure for knowledge dissemination, as well as the division of information collected by the Regional Competence Centers and the National Health Fund, means that this activity (described as S2) may be performed ineffectively.

The lack of specification regarding the exact content of the review (provided by the Regional Competence Centers) significantly limits the advantages related to the fact that an independent governmental institution gives feedback to the hospital assessing the technology (S3).

The vagueness of the model (W1), combined with the lack of central coordination (W4), can result in variability in procedures at both the inter-regional and intra-regional levels. This may lead to inequalities in the application of HB-HTA and increase the significance of threats.

Information about the environment in which the HB-HTA operates comprises not only the HTA framework and the role of the HTA agency, but also the financing scheme and reimbursement of new medical technologies [9,28,29,30]. Many countries have different organizational frameworks of HB-HTA, so there is no uniform approach. For example, no classical HTA agency operates in either Switzerland or in Denmark at the national or regional level [31].

The environment is also important for assessing hospital autonomy in the HB-HTA model. Bottom-up models are highly decentralized, and are based on the assumption that hospitals increase their autonomy by making decisions regarding investment in health technologies, and they receive professional support from the HTA agency [4]. Decentralized models can be viewed by hospital managers as being advantageous because their role in the decision-making process is greater, while also being supported by scientific evidence [12]. On the other hand, centralized HB-HTA models rely on different assumptions about the hospital environment, with a greater role being occupied by central institutions, especially public payers, or governmental institutions, in the decision-making process regarding the assessment of innovative health technologies in hospitals [32].

The proposed model for Poland can be described as being relatively decentralized, as the role of governmental authorities is rather supplementary, yet formally defined—which is perceived as an opportunity for the model’s successful implementation (O1). The compliance of the concept with the current health paradigm and the trends in its change are depicted, for example, in the increasing role of Evidence-Based Management and data analysis (O4). The reduction of health inequalities and the improvement of access to health services should accelerate the dissemination of good practices and thus strengthen the legitimacy of the implementation of the HB-HTA model in Poland.

Compliance with the current health paradigm reduces all of the threats defined in the course of this research, specifically the lack of support for the idea of HB-HTA (T4). HB-HTA should be able to find support from both governmental institutions (e.g., the Ministry of Health with respect to the implementation of results through the public payer system), as well as patients (who should support the popularization of effective new technologies).

The most important threat is the resistance of hospitals (T1), which neutralizes the positive impact of the two most significant opportunities (O2 and O3) and strengthens the impact of the other threats.

Several phenomena seem to influence the adaptiveness of HB-HTA [33]. The first, perceived at the local level, is related to the financing system, the budget constraints, and the policies that organizations must comply with, together with the need to achieve effectiveness and provide high-quality care [34]. The second phenomenon is the increasing awareness of contextual factors, which can significantly influence the adoption of a specific technology [35]. Despite research developments on HB-HTA models, gaps in knowledge are still observable regarding the measurement of models’ adaptiveness, including the tools and methods useful for conducting such assessments.

The adaptiveness should go beyond the environmental characteristics and address institutional bonds, and competitive behavior can also affect the organizational design [36]. The most effective course of action is dependent on both internal and external circumstances. Both internal and external factors can influence organizational design, particularly in terms of how some organizational structural variables are combined, such as the level of the hierarchy, the centralization of decision making, the specialization of labor in the health system, the formalization of processes and procedures, and the personal qualifications of the personnel [33].

The results of the research show that the strongest factor influencing the model’s adaptiveness is the lack of support for the development of HB-HTA (T4). It reduces the possibility of building upon opportunities while also reinforcing all weaknesses, particularly the vagueness of the model (W1) and the lack of central coordination and supervision (W4).

On the other hand, the model’s vagueness can be hindered thanks to the legal and institutional HTA framework in which HB-HTA operates (O1). HTA has a formalized structure in Poland on which the newly defined HB-HTA process can be based, thereby limiting (or even eliminating) the negative consequences of the vagueness in the primary concept.

The last dimension is the responsiveness of the HB-HTA model. One could argue that the greatest responsiveness could be achieved by engaging a variety of stakeholders in the governance model [37]. Given Kooiman’s definition of governance as a pattern of conduct or structure that emerges in a socio-political system as a result or consequence of the interactive intervention efforts of all of its active participants [38]. This understanding of governance leads to the maintenance and cultivation of the relationship between society and decision makers. In political sciences, responsiveness refers to the relationship between the desires and demands of citizens, and the decisions that directly or indirectly affect them, as well as the ability of politicians to collect responses and address the current preferences and sentiments of the members of a society.

The process of introducing HB-HTA in Poland has followed the assumptions of the pragmatic model of public responsiveness by combining social expectations with the knowledge of health experts. Our research indicates that the most important, here, are two features: organizational and procedural vagueness (W1), and a lack of central coordination and supervision (W4).

Any HB-HTA model that lacks a precise definition of the role of all stakeholders, as well as the stages of the process and their products, limits the chances of the successful implementation of the health technology. This further reinforces the reluctance of both hospitals (T1) and decision makers (T4).

The lack of central coordination is equally significant. It hinders the chances of disseminating good practice and reduces the possibility of utilizing existing managerial competencies. The heterogeneous nature of the process will discourage specialists from engaging in the new area, which could be perceived as being risky, or not systematized. The lack of coordination reinforces all of the threats, in particular the problem with data transmission and quality (T3).

## 5. Conclusions

When designing an HB-HTA model for hospitals operating in a specific country or region, it is important to subject it to a comprehensive assessment. There is no pattern that is equally applicable regardless of the local healthcare context [27]. For this reason, a methodology for assessing the HB-HTA solution should be sought. The one proposed in this study, and presented on the basis of the example of the HB-HTA model proposed for Poland, is Dynamic SWOT analysis.

An extension of the classical analysis of strengths, weaknesses, opportunities and threats, the Dynamic SWOT was proposed as an evaluation tool. It enables researchers to capture the dynamics of the model by analyzing the influence of individual factors on (1) the characteristics of the HB-HTA model; (2) the environment in which it operates; (3) the adaptiveness of the model to the environment; and (4) the model’s responsiveness. This approach makes it possible to assess the model with respect to the risks associated with the regional or national healthcare system in which the HB-HTA is to be performed and, if necessary, adjust its parameters in order to maximize its effectiveness and efficiency.

The Dynamic SWOT analysis can be used to assess any public policy concept, model, or framework. Its results are subject to the primary identification of strengths, weaknesses, opportunities, and threats, as well as the further determination of their impact. The evaluation process will be more objective when performed with reference to good practices described in the literature—as is the case here with the AdHopHTA guiding principles. Additionally, the assessment can be conducted by a panel of experts discussing their individual results.

The aim of this study was to assess an exemplary HB-HTA model against good practices, and to analyze its responsiveness and adaptability. The HB-HTA model was parametrized, and a classical SWOT analysis was performed. Specific features were further categorized according to the AdHopHTA guiding principles in order to identify areas that require further refinement. The conclusions from the Dynamic SWOT analysis indicate that implementing the model in the shape initially designed may hinder its strengths and reinforce its weaknesses.

The strengths are hindered primarily by two factors: the opposing direction of the model’s weaknesses, and the impact of existing threats. The weaknesses are reinforced by their mutual connections with each other (thus increasing their effect), as well as by the existing threats. One way to improve the chance of the model being implemented is to eliminate the model’s vagueness. Furthermore, depending on how the model is to be performed, it should potentially undergo another Dynamic SWOT analysis, as the model’s internal features and their interrelationships may change further.

The D-SWOT analysis tool has several limitations. Identifying the factors and classifying them can be challenging, particularly as a factor can be both a strength and a weakness. When neglected, strengths can become weaknesses, and missed opportunities can become threats. Both the classification of factors and the assigning of weights to them are subject to the individual assessment of the researchers.

Cognizant of the method’s limitations, the authors tried to minimize their impact by carrying out work individually, comparing the results, and discussing discrepancies during cyclical meetings, among other things.

The results of this research have led to the HB-HTA model being modified in order to remove the vagueness. At the same time, the proper policies have been put in place to address the threats [39].

## Figures and Tables

**Figure 1 ijerph-19-07281-f001:**
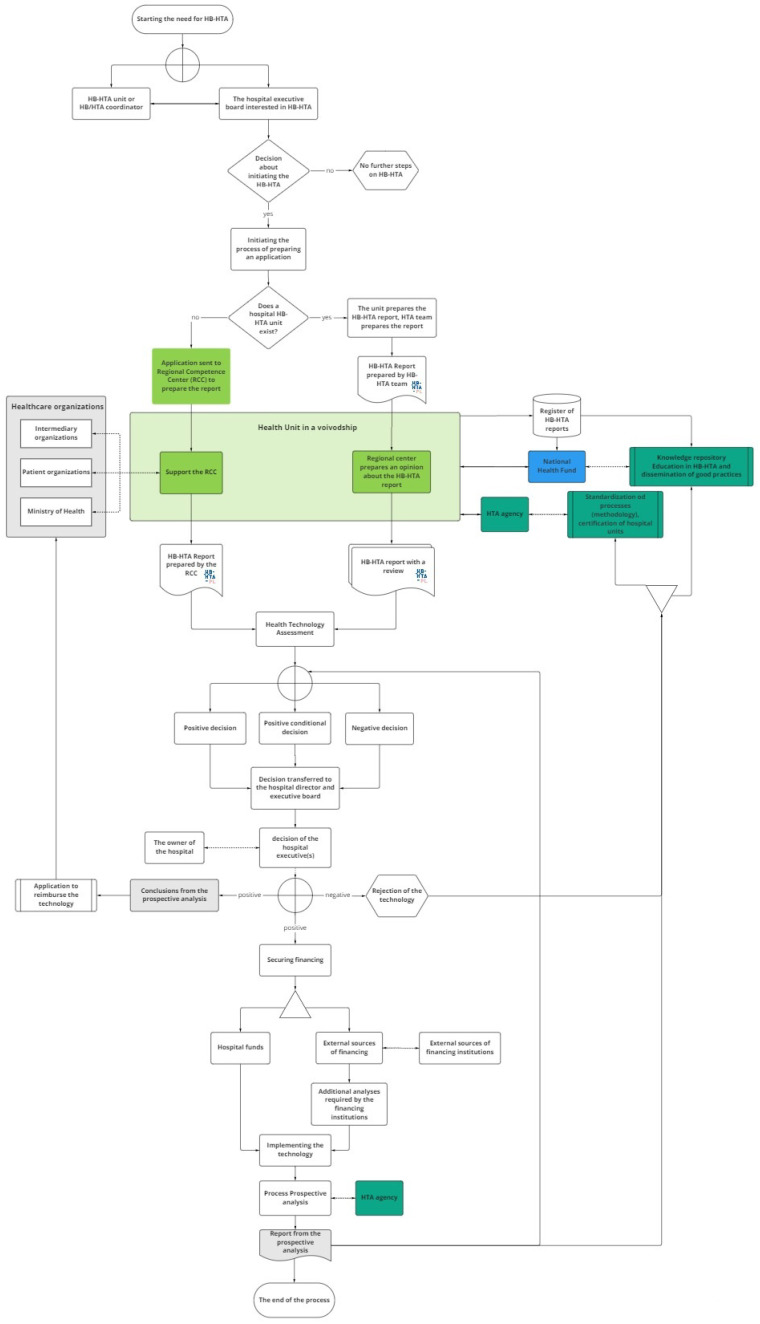
The model designed for Poland. Source: [24].

**Figure 2 ijerph-19-07281-f002:**
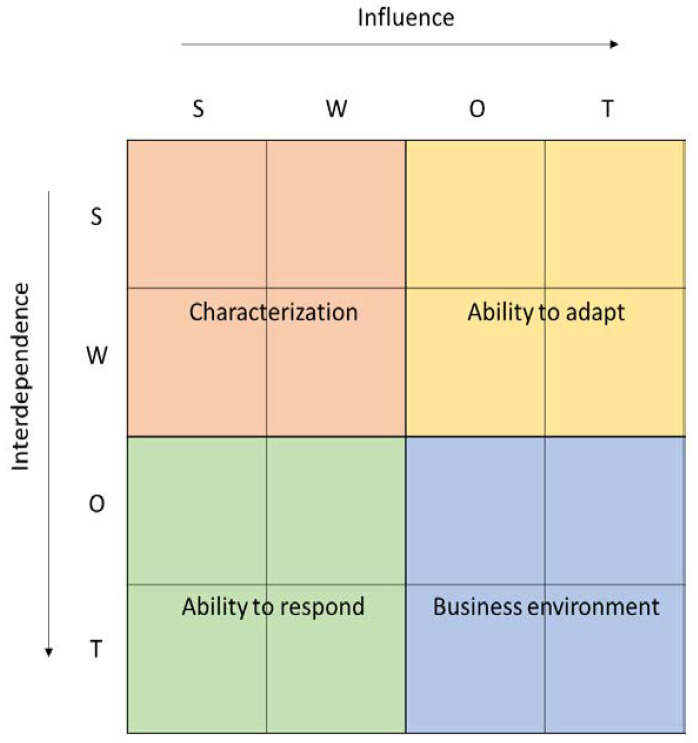
Dynamic SWOT matrix. Source: [16].

**Figure 3 ijerph-19-07281-f003:**
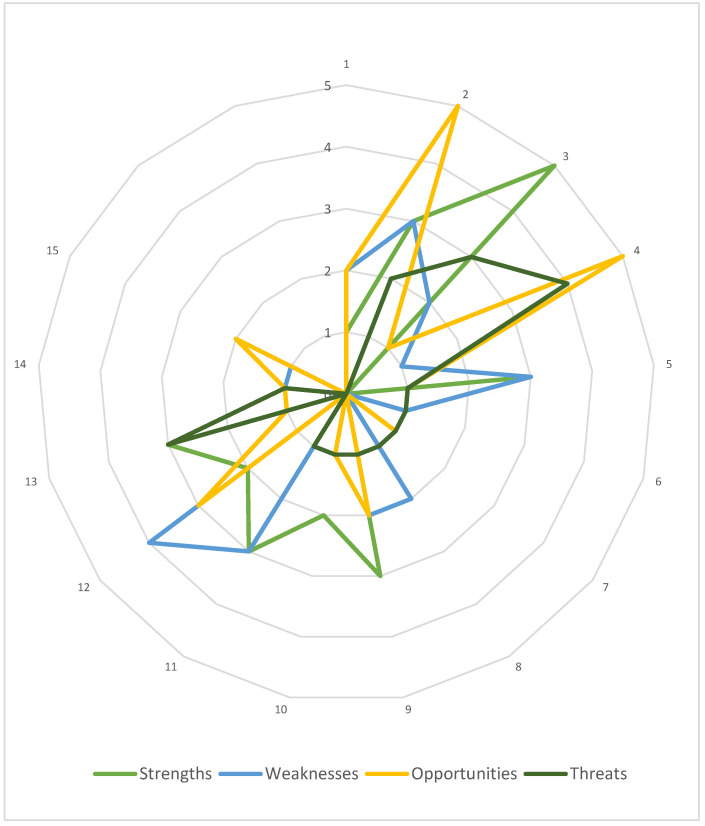
The Polish HB-HTA model with reference to the AdHopHTA guiding principles. Note: numbers 1–15 denote each guiding principle as defined in Table 2. Source: own work.

**Figure 4 ijerph-19-07281-f004:**
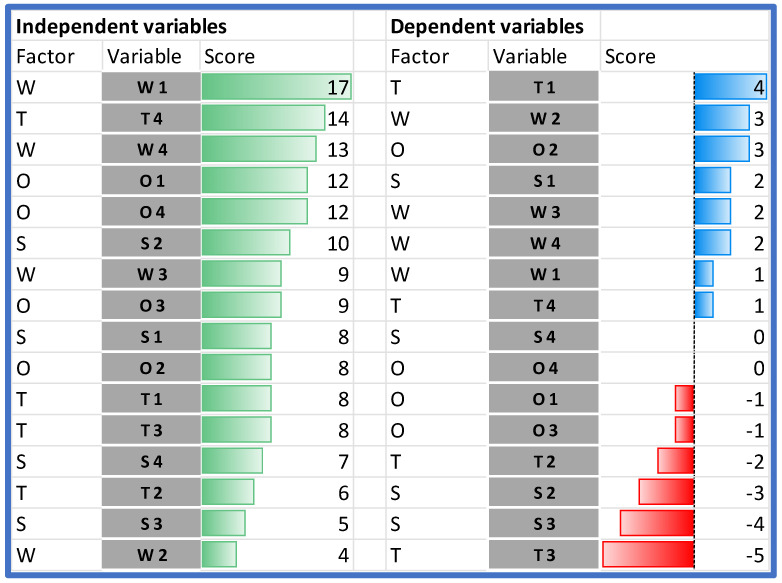
Dependent and independent variables in the order of their impact. Source: own work.

**Figure 5 ijerph-19-07281-f005:**
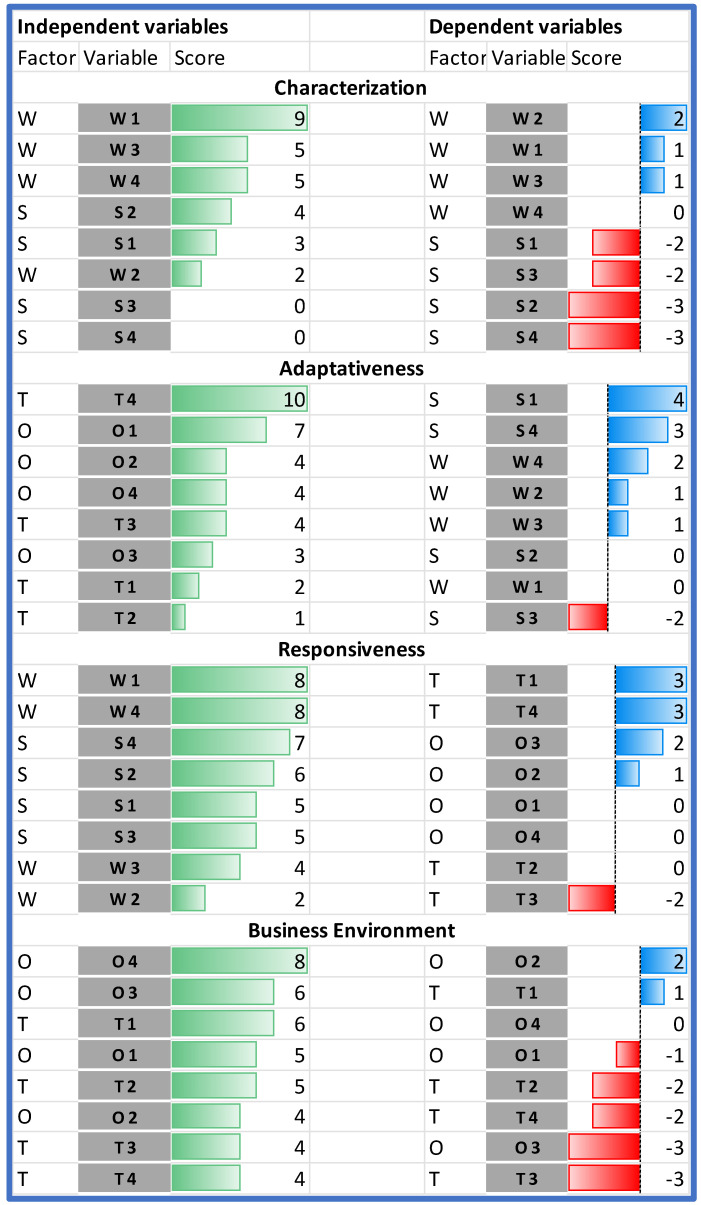
Characterization of adaptiveness, responsiveness, and business environment in the HB-HTA model. Source: own work.

**Table 1 ijerph-19-07281-t001:** Roles in the HB-HTA process. Source: [24].

**Hospital** ⇩ **RCC** ⇩ **NHF** ⇩ **HTA Agency**	‹ The initiator of the HB-HTA process ‹ The author of the HB-HTA report
	‹ The coordinator of the HB-HTA model implementation at the regional level ‹ The provider of the review on the HB-HTA report
	‹ The organizer of the regional health information database ‹ The HB-HTA model disseminator and educator
	‹ The author of the HB-HTA methodology ‹ The certificate provider for the HB-HTA hospital units

**Table 2 ijerph-19-07281-t002:** HB-HTA guiding principles.

Dimension	No.	Guiding Principles
THE ASSESSMENT PROCESS	1	HB-HTA report: scope, hospital context, and informational needs
	2	HB-HTA report: methods, tools, and transferability
	3	HB-HTA process: independent, unbiased, and transparent with stakeholder involvement and communication
LEADERSHIP, STRATEGY AND PARTNERSHIPS	4	Mission, vision and values, and governance
	5	Leadership and communication policy/strategy
	6	Selection and prioritization criteria
	7	Process of disinvestment
	8	Improving through innovation
	9	Knowledge and resource sharing
	10	Collaboration with HTA organizations
	11	Links with allies and partners
RESOURCES	12	Skilled human resources and career development
	13	Sufficient resources (including financing) *
IMPACT	14	Measuring short- and medium-term impact
	15	Measuring long-term impact

* The authors changed the definition of the guiding principle initially defined as ‘Financial resources’. In the authors’ opinion, the division of the resources dimension into human and financial resources only is not sufficient. Source: own work based on [2] (p. 102).

**Table 3 ijerph-19-07281-t003:** SWOT analysis.

**Strengths**	**Weaknesses**
Uniform methodology at the country level (S1) Knowledge database and unified knowledge dissemination procedure (S2) Review by an independent regional governmental institution (S3) Independence of hospitals with the formal institutional support (S4)	Organizational and procedural vagueness (W1) The need to provide resources in several institutions in parallel (W2) The dispersion of the process between institutions (W3) Lack of centrally conducted coordination and comprehensive supervision (W4)
**Opportunities**	**Threats**
Legal and institutional framework of the HTA process (O1) Existing increasing managerial competencies in hospitals and entities responsible for the coordination of the process (O2) Dissemination of HB-HTA good practices (O3) Compliance with the current health paradigm and trends in its change (O4)	Resistance of hospitals against the introduction of changes and implementation of external regulations in the field of HB-HTA (T1) Monopolization of the HB-HTA process within public structures (T2) Problem with data transmission and quality (T3) No support for the development of HB-HTA (T4)

Source: own work.

**Table 4 ijerph-19-07281-t004:** Results of the Dynamic SWOT analysis.

	S1	S2	S3	S4	W1	W2	W3	W4	O1	O2	O3	O4	T1	T2	T3	T4	
S1	--	1	0	0	−2	0	0	−1	1	2	1	2	0	0	−1	−1	2
S2	0	--	0	0	−2	0	−1	0	1	1	0	1	−1	0	−1	−1	−3
S3	1	0	--	0	−2	0	0	−1	1	0	0	0	−1	0	−1	−1	−4
S4	1	1	0	--	−2	−1	−1	−1	1	1	1	1	0	0	0	−1	0
W1	0	0	0	0	--	0	1	0	−2	0	0	0	0	0	0	2	1
W2	0	0	0	0	1	--	0	1	−1	0	0	0	0	1	0	1	3
W3	0	−1	0	0	0	1	--	1	0	0	0	0	0	0	0	1	2
W4	−1	−1	0	0	0	0	2	--	0	0	−1	0	0	0	1	2	2
O1	0	0	0	0	0	0	0	0	--	0	0	0	0	0	0	−1	−1
O2	0	1	1	1	0	0	−1	−1	1	--	1	1	−1	0	0	0	3
O3	2	2	1	1	−2	1	−1	−2	1	1	--	2	−2	−2	−1	−2	−1
O4	0	0	0	0	0	0	0	0	0	0	0	--	0	0	0	0	0
T1	1	−1	1	−2	2	0	1	1	−1	−1	−1	−1	--	2	2	1	4
T2	0	−1	0	−1	1	0	0	1	−1	0	−1	−1	1	--	0	0	−2
T3	−2	−1	−1	−1	1	0	0	2	0	−1	−2	−1	1	0	--	0	−5
T4	0	0	−1	−1	2	1	1	1	−1	−1	−1	−2	1	1	1	--	1
	8	10	5	7	17	4	9	13	12	8	9	12	8	6	8	14	

Source: own work.

## Data Availability

Not applicable.

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
