# Peer review of "Using the Dynamic SWOT Analysis to Assess Options for Implementing the HB-HTA Model"

_ijerph, 2022, doi:10.3390/ijerph19127281_

Round 1

Reviewer 1 Report

The paper describes an ambitious and interesting topic related to the importance of implementing innovations in healthcare organisations, by making an excursus a tool is proposed to investigate the factors that can foster the successful implementation of the HB-HTA ecosystem. In order to pursue this goal, the authors use a SWOT analysis method to create an HB-HTA model best suited to the regional or national healthcare system.
The paper should therefore be further investigated in order to build the model to which it aspires. The form of the text is not scientific but rather journalistic, often resorting to the plural first person instead of remaining within the scientific nature of the research (as line 59). The three questions posed in the introductory section (lines 63-67 are not fully answered in the text). Furthermore, the content concerning the structure of the paper is missing in the introductory section. In particular, it should be specified how a specific case referred to Poland can then be adopted in other contexts to support what is stated in the abstract. Otherwise, if this were not the case, it would not represent an advancement in the state of the art of the literature in general as stated, but only in that of the country of Poland, and this should therefore also be stated in the title.
In the section on materials and methods from lines 94-96, in which the authors explain how the research is articulated, it should be implemented, contextualised and argued, and whether this is the appropriate section in which to insert this aside. The methodology is trivialised and poorly articulated; it should be improved and a clarifying methodological outline should be added. Also, the HB-HTA model should be better explained. In section two there is no clear distinction between the decision makers and stakeholders involved, on the consistency and image 1 is totally illegible. It is not clear from lines 161 to 170 whether the research he is talking about is done by the authors or by the team, in the second case it should be made clear what the authors' role in the research is and why it is being talked about. The source at line 175 and the captions are not appropriately reported according to the journal's editorial standards, so the authors are requested to adjust them. From line 181 to 190 you need to argue what you wrote and contextualise it by explaining the objective, the text is unclear. At line 199 a space must be inserted between the numbers indicating the range. At line 222 "All factors in SWOT analysis are simultaneously dependent and independent variables, which reflects the dynamics of the model." It does not have a clear or scientific meaning like many other parts of the text and therefore the authors are invited to better explain what is written by them. In the last part of the second section, reference is made to a matrix that is not present, which should be given in full, helping readers to understand it.
In section three, the pictures and captions should follow what was already indicated in section two by aligning them with the editorial standards of the magazine, as should all those in the text. Furthermore, the figure inserted in section three about the model is not explained or argued but simply presented without scientific support to what is graphically rendered. In Table 3, the alphanumeric codes are not readable, especially those in the first column. At line 360, at line 396, at line 429, at line 545, if bold is not necessary please do not use it and see the editorial standards of the journal. In particular, section 4 should state the potential transferability uses of the model in other contexts. The conclusions do not correspond to what is stated in the abstract and introduction, nor to the research questions stated at the beginning. 
The contribution requires a revision of the English language and also of the entire structure as well as most of the content, and does not return the research in a comprehensive manner by making methodological leaps and gaps in terms of content. In order for it to be published in this journal, it requires a complete deepening and transformation of the paper. Finally, the author contributions were not included.

Author Response

Dear Reviewer,

thank you for taking the time and reviewing our article. We followed all your suggestions and addressed all the issues included in the review. I am attaching a table with all details about the changes made in the manuscript. Should you have any questions, we'd be delighted to answer them. 

Yours sincerely, 

Katarzyna Byszek on behalf of the authors

Reviewer 2 Report

The article aims to contribute to the policy debate on the effective model of hospital based health technology assessment.

HB-HTA can facilitate bringing together evidence and other relevant and reliable information

for hospital managers and other decision-makers and lead them to good investment decisions.

They formulated three research questions to assess the HB-HTA model.

The paper introduces a dynamic assessment approach for answering the research questions and contributes to the literature discussion twofold. Firstly, a conceptual frame work is proposed to assess features of an HB-HTA model designed for a country or region, around the concepts of adaptiveness and responsiveness to features of a healthcare system that is present there.

Secondly, a Dynamic Strengths, Weaknesses, Opportunities, and Threats (SWOT) analysis framework has been used to predict the results of an implementation of models in public policy, and in this case – the HB-HTA implementation in response to the dynamics of the healthcare system. The Dynamic SWOT framework provides evidence of the interde pendency between the strengths and weaknesses of the HB-HTA model and opportunities and threats resulting from the system-based conditions.

They drew on a study of Gałązka-Sobotka and Kowalska-Bobko on the HB-HTA model designed for Poland, analyzing its features with reference to AdHopHTA good practices and the Polish healthcare system context. They described the model based on two sets of variables – endogenous factors that present how well the proposed model fits into the AdHopHTA framework and exogenous factors based on potential risks associated with the local system-based context.

A broader introduction and summary are missing.

Author Response

Dear Reviewer,

thank you for taking the time and reviewing the article. We addressed your suggestion and made changes to the introduction and the description of materials and the methods used to assess the HB-HTA model, as well as in the part with conclusions. We believe that your review contributed to the greater quality of the article.

Yours sincerely,

the authors

Reviewer 3 Report

see file

Author Response

Dear Reviewer,

thank you for taking the time and reviewing our article. Your suggestions and comments were very helpful and we included all of the in the manuscript:

  1. we added figure no. 3 to better explain the method used for the data analysis;
  2. we unified the format of references
  3. we read and included in the part with conclusions one of the findings from the article that you recommended.

We believe that your review helped us to improve the quality of the article and we are grateful for this.

Yours sincerely,

the authors

Round 2

Reviewer 1 Report

the document is considerably improved, in order to make it publishable, the following aspects need to be further modified: 

- image 1 must be rendered with better quality and if this is not possible, it must be redone as there are formats and dpi to be observed according to the editorial standards of the magazine. 

- the introductory part must be completely improved in the light of the latest data on the sector, which includes a more recent date, as well as the arrangement and explanation of the text that will follow in the respective sections and contents. 

- revision of the English and bibliography is recommended. 

Author Response

Dear Reviewer,

thank you again for taking the time and reviewing our article. We have made the following changes:

  1. changed Figure No. 1 - we hope it is readable now and meets the quality requirements
  2. modifications in the introduction - we added more information on current developments in HB-HTA and regulations, along with the description of the contents
  3. we revised the article, including the bibliography.

Should you have any further questions or concerns, we will be happy to address them.

Yours sincerely,

the authors
